

# Measurements on physical snow properties in Dronning Maud Land, Antarctica

Leena Leppänen[1,2], Antero Kukko[3], Aleksi Rimali[1], Aku Riihelä[4], Priit Tisler[4]

[1]Finnish Meteorological Institute, 99600 Sodankylä, Finland
[2]Arctic Centre, University of Lapland, 96100 Rovaniemi, Finland
[3]Finnish Geospatial Research Institute, 02150 Espoo, Finland
[4] Finnish Meteorological Institute, 00101 Helsinki, Finland

*Correspondence to*: Leena Leppänen (leena.leppanen@fmi.fi)

**Abstract.** Understanding the seasonal evolution of Antarctic snow is essential for interpreting satellite observations and quantifying surface mass balance in Antarctica. In this study, we present in situ measurements collected during the Finnish Antarctic Research Programme (FINNARP) 2022 expedition at the Finnish Aboa station, located in Dronning Maud Land,
Western Antarctica, which is characterized by snow accumulation. Field observations were carried out weekly at the AWS5 automatic weather station, situated approximately 10 km from Aboa, as well as at selected overpass locations of the IceSat-2 and CryoSat-2 satellites. This manuscript presents data from the AWS5 site, where seasonal snow evolution was systematically monitored. The measurements included continuous meteorological observations from the weather station maintained by FINNARP, detailed snow pit profiles, ground-based and drone-based radiation measurements, and snow
surface roughness observed using both drone-mounted and backpack-mounted laser scanners. Drone-based measurements enabled spatially extensive coverage using a laser scanner, a hyperspectral camera, and a pyranometer. Spatial variability of snow surface properties was assessed at five locations in addition to the primary snow pit. The collected dataset provides valuable insight for improving satellite remote sensing and for advancing our understanding of the relationships between albedo, surface roughness, and physical snow properties.

**1 Introduction**

The paper describes a data set collected in 2022-2023 in Antarctica as part of the Low orbit altimetry, albedo, and Antarctic Snow and Sea-ice Surface Roughness (LAS3R) project funded by the Research Council of Finland. Measurements were conducted during Finnish Antarctic Research Programme (FINNARP) 2022 expedition to Finnish Aboa station located in Dronning Maud Land in western Antarctica. Aboa is located in the snow accumulation area (Isaksson and Karlen, 1994;
Reijmer and Broeke, 2003). The field measurements were made weekly at AWS5 weather station (−73.105034S,



−13.162370W), approximately 10 km from Aboa station, and at different satellite overpass locations on pre-checked GPS routes typically 5-35 km from the station. The presented data in this manuscript are from the AWS5 site, where seasonal snow evolution was monitored. Measurements included automated meteorological measurements at the weather station, snow pit measurements, radiation observations on the ground, radiation observations on a drone, and snow surface roughness observed with a drone-based laser scanner and a backpack-based laser scanner. In total eight measurement occasions were made at AWS5 site and 16 measurements at different IceSat-2 and CryoSat-2 satellite overpass locations, including measurements at Riiser-Larsen shelf-ice approximately 100-130 km north-west from the Aboa station.

Majority of the earlier snow pit studies around Aboa station have been longer transects including only few measurements at vicinity of the Basen nunatak without repetition at the same location (Isakson and Karlene, 1994; Kärkäs et al., 2002; Kärkäs et al., 2005; Rasmus, 2006; Ingvander et al., 2016; Vihma et al., 2011; Järvinen and Leppäranta, 2013). However, studies in 2006, 2007, 2009 and 2014 had a stable snow measurement site close to Basen (-73.064283S, -13.47668W) where snow pits were made 1-2 times per day (Vihma et al., 2011; Pirazzini et al., 2015). In 2018, almost daily snow pit measurements were made at the AWS5 site (FINNARP, 2019). Previous unpublished drone measurements near Basen nunatak have been related to albedo, atmospheric and meteorological observations.

Isakson and Karlene (1994) studied snow accumulation with a stake transect from coast to Heimefrontfjella, and with density measurements in snow pits nearby. They found that the spatial variations in accumulation are large over short distances, local topography and slope changes cause smaller-scale variations in accumulation, and a decreasing accumulation rate from the coast and up to 300 km inland is visible in the stake measurements. Reijmer and Broeke (2003) studied surface mass balance derived from automatic weather stations, including AWS5 station close to Basen nunatak. They summarized that at AWS5 the accumulation is strongest in autumn and winter, with 65-75 % of the annual accumulation occurring in these seasons. The autumn or winter maximums differ significantly from the spring or summer minimums. The mean accumulation at AWS5 was 17.7 cm ±3.6 cm in their study. They also say that the tendency of larger events contributing (positively) to the total accumulation is a function of distance from the coast, elevation and annual accumulation.




**Table 1: Earlier snow pit measurements around Aboa station**

| Year | Measured parameters | Num. of pits | Location | Reference |
|------|---------------------|--------------|----------|-----------|
| 1988-1989 | Snow accumulation, density | 529 | Transect from coast to Heimefrontfjella | Isakson and Karlene, 1994 |
| 1999-2000 | Temperature, density, grain-size and shape, dielectric constant, wetness, electrical conductivity, number and position of ice layers, albedo and solar-radiation attenuation. Oxygen isotope ratio ($\delta^{18}$O) at five sites. | 17 | Transect from coast to Heimefrontfjella | Kärkäs et al., 2002; Kärkäs et al., 2005 |
| 1999-2000 | Stratigraphy and density | 2 | Together with GPR measurements close to Aboa | Sinisalo et al. (2003) |
| 2000-2001 | Visible stratigraphy, temperature, density, grain size and shape, dielectric constant, wetness, conductivity, pH and oxygen isotope ratio ($\delta$18 O). | 11 | Transect from coast to Heimefrontfjella | Kärkäs et al., 2005 |
| 2003-2004 | Visible stratigraphy, temperature, density, grain size and shape, dielectric constant, wetness, conductivity, pH and oxygen isotope ratio ($\delta$18 O). | 10 | Transect from coast to Heimefrontfjella | Kärkäs et al., 2005 |
| 2003-2004 | Density and layer hardness (hardness not measured for all pits) | 78 | 310 km long transect from coast to Svea | Vihma et al., 2011 |
| 2004-2005 | Density, layer hardness and temperature | 70 | 310 km long transect from coast to Svea | Vihma et al., 2011 |
| 2006-2007 | Temperature, density, short and longwave radiation | 45 | Snow site close to Aboa | Vihma et al., 2011 |
| 2007-2008 | Particle size and shape, density, conductivity, hardness, surface morphology | 62 | Transect from coast to plateau | Ingvander et al., 2016 |
| 2007-2008 | Temperature, density, short and longwave radiation | 32 | Snow site close to Aboa | Vihma et al., 2011 |
| 2009-2010 | Temperature, density, grain size (from photographs of single grains) and spectral reflectance | 8 | Snow site close to Aboa | Pirazzini et al., 2015 |
| 2009-2010 | Thickness, density, hardness (hand test), liquid-water content, and grain size and shape (from photographs of grains) | 8 | Transect from Aboa to Svea | Järvinen and Leppäranta, 2013 |
| 2014-2015 | Profiles of density and temperature, grain size (photographs from single grains), stratigraphy with hardness and wetness, profile photo, and spectral reflectance | 43 | Snow site close to Aboa | |
| 2018-2019 | Profiles of temperature and density, stratigraphy, penetration force with SnowMicroPen | 1-2 almost daily | AWS5 | FINNARP, 2019 |



Sinisalo et al. (2003) measured snow layers with ground penetrating radar during FINNARP 1999 from the topmost 50 m of
the snowpack. They also compared radar observations with a 10 m deep snow pit and a 20 m deep snow pit. However, only
the highest frequency, 800 MHz, gave results from the topmost 1–2 m. Moreover, fine layer structure was not visible in the
radar data.

Snow pits with depth of 1–2 m were made from coast to mountain range of Heimefrontfjella, including 17 snow pits in
FINNARP 1999, 11 snow pits in FINNARP 2000 and 10 snow pits FINNARP 2003 expeditions (Kärkäs et al., 2002; Kärkäs
et al., 2005). They concluded that mean grain size in the annual layer ranged between 1.5–1.8 mm, so that grain size
decreases exponentially with the distance from the ice edge. They found that the snow cover contains numerous thin ice
layers and crusts (0.5–2 mm), which, according to Goodwin (1991), form at the surface in summer due to strong incident
solar radiation and wind. According to Mosley-Thompson et al. (1985), a high-density layer with fine grains forms in winter,
while summer layers have lower density and bigger grains. Near the Basen nunatak from 20 cm depth, well-rounded grains
represent the most common grain shape (Kärkäs et al., 2002). Snow grain size was measured in a transect from coast to
plateau in 2007-2008 (Ingvander et al., 2016). They concluded that there are significant differences in particle size between
the coast and the plateau, and the range in snow particle size varies significantly between the geographical regions at local
scale due to different surface conditions such as microtopography. According to Kärkäs et al., (2002), daily temperature
variations were significant down to about 30 cm depth. During the FINNARP 1999 expedition the mean density in the first
metre with the standard deviation was $409 \pm 8$ kgm$^{-3}$ (Kärkäs et al., 2002).

FINNARP 2006 and 2007 expeditions made daily snow pits (depth 50-90 cm) close to Basen nunatak including temperature
and density measurements (Vihma et al., 2011). They resulted large inter-annual variations in the uppermost 50 cm of the
snowpack; In 2007–2008 the snow temperatures were as much as 1.4 °C lower than in the previous summer; In 2006–2007
the snow density was higher than in 2007–2008 in the whole 50 cm layer, the maximum difference of 40 % at the surface.
Also, they concluded that considering time scales of hours and days, the temperature of the uppermost 20 cm of the
snowpack was strongly correlated with the mean air temperature during the preceding 6 to 12 h. At deeper layers the most
important time scale was 72 h. They deduced that short-term temporal variations of snow density were less well explained,
but the history of solar radiation on a time scale of 10 days controlled the density at the depth of 30 cm. They also observed
that temperature maxima were close to the melting point during most clear and cloudy days, but temperature minima were
much lower during clear days than during cloudy days.

Spectral and total albedo measurements were made along a line from the coast via Aboa to Svea station during FINNARP
1999 expedition (Rasmus, 2006; Kärkäs et al., 2002). They concluded that mean spectral albedo for snow from the Aboa
traverse showed values of 0.95 in the visible band with minor wavelength dependence. After 700 nm, the wavelength

dependence increased and at 1034 nm, the albedo dipped down to 0.70. They observed that the variations due to ambient conditions seem to outweigh the spatial variations. They showed that overcast and partly cloudy albedo cases resulted values that were 0.1 higher, between 400 and 700 nm, than those for the clear sky or almost clear sky cases. At longer wavelengths, the partly cloudy cases showed slightly lower values than those of the overcast cases. Midday total albedo values were between 0.85 and 0.90 for overcast conditions. Some diurnal variations were found with the lowest albedos being in the mornings and in the evenings.

Solar radiation between 400-900 nm and snow pits were measured in the FINNARP 2009 expedition to study transmittance through snow (Järvinen and Leppäranta, 2013). Then a total of eight pits were made between Aboa and Svea, including measurements of thickness, density, hardness (hand test), liquid-water content, and grain size and shape (from photographs of grains). They concluded that the predominant grain type was large and rounded particles, and the predominant grain size was 1 mm in every snow pit. Snow pits including temperature, density, grain size and spectral albedo were made on eight days close to Basen nunatak at FINNARP 2009 expedition for comparison between measured and modelled albedo (Pirazzini et al., 2015). Stratigraphy observations were only made at a qualitative level, without snow hardness measurements and systematic recording of layer properties.

Aim of this paper is to describe the data set collected during the FINNARP 2022 expedition and provide example data from the AWS5 site. All measurement methods used in the field campaign are described with details and example data for the AWS5 field site is presented. In the end, short conclusions are given.

## 2 Field measurements

Field measurements were conducted in CryoSat-2 and IceSat-2 satellite overpass locations at the vicinity of Aboa station as well as repeatedly at the AWS5 site (Figs. 1 and 2) between November 2022 and February 2023. Total 26 measurement occasions were conducted. Measurements at the IceSat-2 and CryoSat-2 satellite overpass locations were made for calibration and comparison of these data with simultaneous in situ data (within a few hours, if not simultaneous). To follow the evolution of snow properties over the summer 2022-2023, repeated measurements in the vicinity of the AWS5 weather station are studied. Measurement configurations and coordinates are presented in Table 2.





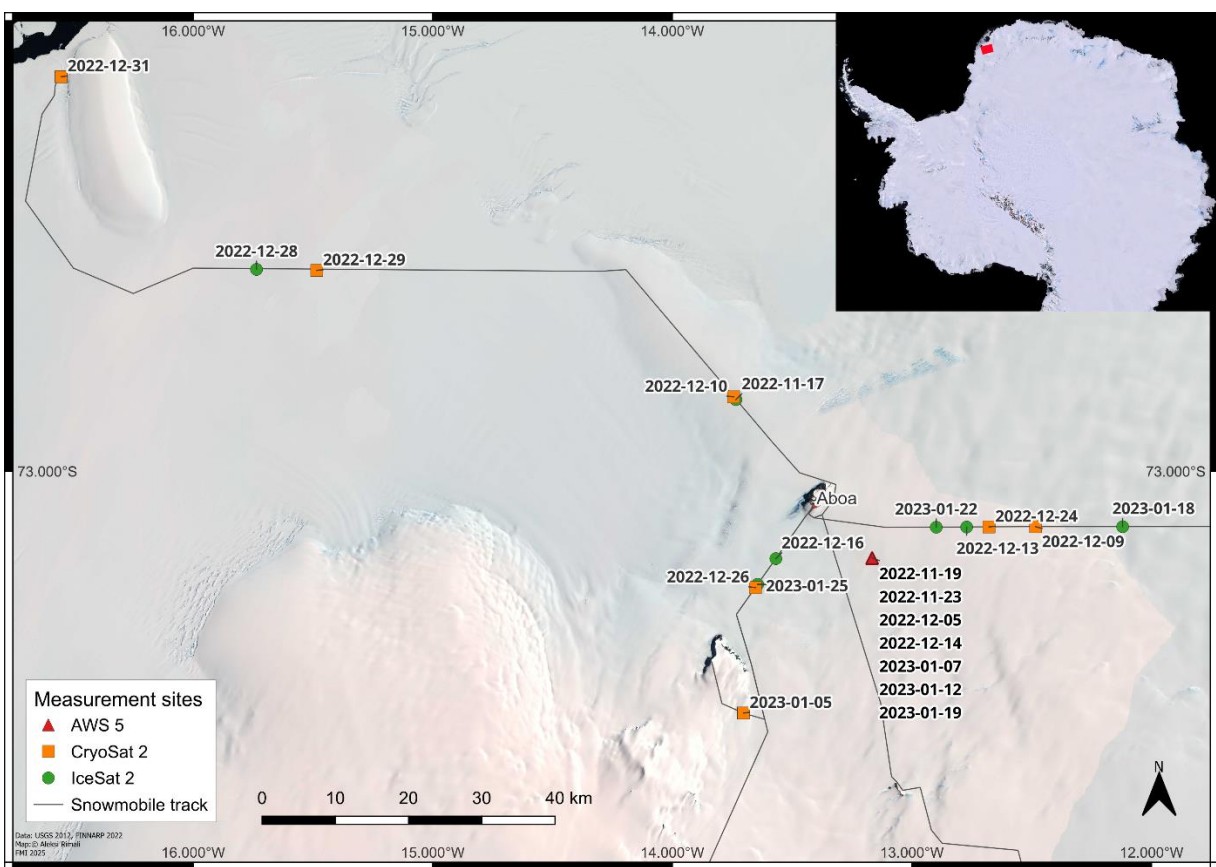

Figure 1: Map of the area where field sites are located. Dates indicate when measurements were made.

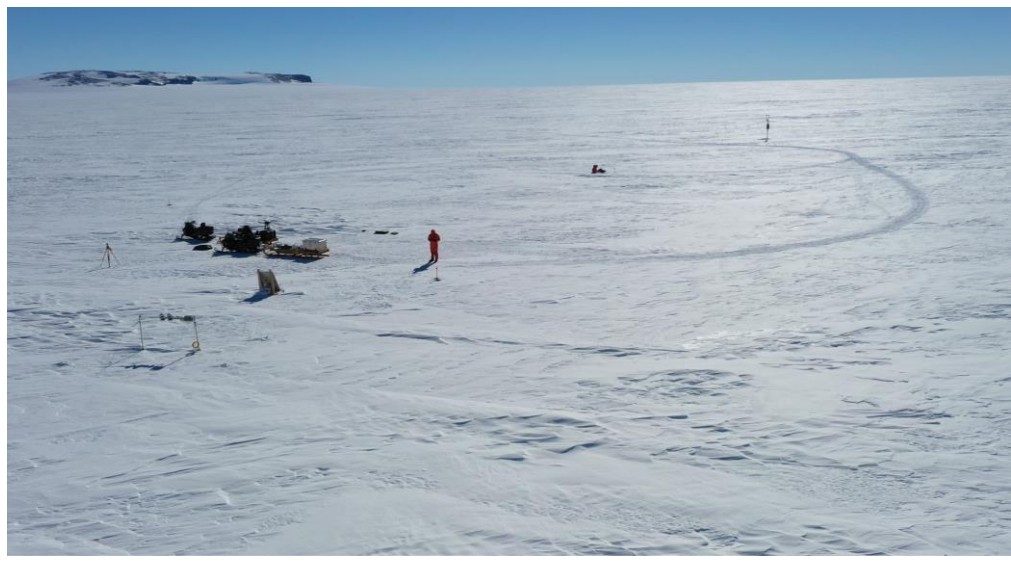

Figure 2: Overview of AWS5 measurement site from a drone.



**Table 2: Date and conducted measurements for each measurement occasion. Cloudiness is also noted. Measurements at the AWS5 site are shaded with grey.**

| Date | Snow pit | Laser scanner drone/ backpack | Hyperspectral camera (amount of image locations) | CM11 | Coordinates | Clouds |
|---|---|---|---|---|---|---|
| 17.11.2022 | x | - | 84 | x | −72.912162S, −13.734135W | 0/8 |
| 19.11.2022 | - | Drone | 36 | x | −73.105034S, −13.162370W | 1/8 |
| 23.11.2022 | x | Drone | 36 | x | −73.105034S, −13.162370W | 1/8 |
| 29.11.2022 | x | Drone | - | - | −73.105034S, −13.162370W | 6/8 |
| 5.12.2022 | x | Drone | 36 | x | −73.105034S, −13.162370W | 3-4/8 |
| 9.12.2022 | x | Drone | 80 | x | −73.067404S, −12.483155W | 8/8 |
| 10.12.2022 | x | Drone | - | - | −72.908774S, −13.743972W | 7-8/8 |
| 13.12.2022 | x | Backpack | 98 | x | −73.067902S, −12.771037W | 1/8 |
| 14.12.2022 | x | Backpack | 36 | x | −73.105034S, −13.162370W | 5-6/8 |
| 16.12.2022 | x | Backpack | 112 | x | −73.106220S, −13.567608W | 6-7/8 |
| 24.12.2022 | x | Drone | 112 | x | −73,067715S, −12.677909W | 8/8 |
| 26.12.2022 | x | Drone | 98 | x | −73.141305S, −13.652969W | 7/8 |
| 28.12.2022 | x | Drone | 112 | x | −72,751604S, −15,735542W | 4-6/8 |
| 29.12.2022 | x | Drone | 112 | x | −72.752794S, −15.483951W | 8/8 |
| 31.12.2022 | x | Drone | 84 | x | −72.511447S, −16.550886W | 5-7/8 |
| 3.1.2023 | x | Drone | - | x | −73.068080S, −12.893071W | 1/8 |
| 5.1.2023 | x | Drone | 140 | partly | −73.292645S, −13.702868W | 0/8 |
| 7.1.2023 | x | Drone | 72 | x | −73.105034S, −13.162370W | 8/8 |
| 12.1.2023 | x | Drone | 72 | x | −73.105034S, −13.162370W | 1/8 |
| 14.1.2023 | partly | Partly with drone | 84 | - | −73.061334S, −13.288573W | 2-5/8 |
| 18.1.2023 | x | Drone | 134 | x | −72.834869S, −13.957230W | 1-2/8 |
| 19.1.2023 | x | Drone | 66 | x | −73.105034S, −13.162370W | 8/8 |
| 22.1.2023 | x | Drone | 112 | x | −73.068086S, −12.898161W | 5-7/8 |
| 25.1.2023 | x | Drone | 42 (partly) | x | −73.137341S, −13.643303W | 0/8 |



### 2.1 Avartek Boxer drone

Avartek Boxer is a Finnish made octocopter (Fig. 3a), which is operated electrically, but it has a gasoline generator to
produce electric power during flight. This drone was used for measurements with a laser scanner. The maximum take-off
weight of the UAV is 27 kg. The two fuel tanks on both sides of the UAV take altogether seven litres of gasoline (~5 kg),
and based on the experiences in this campaign the fuel consumption was about 2.2-2.5 l/h. Consumption depends on the
prevailing wind conditions, and 12 m/s wind speed was kept as a safety limit for the flights.

We conducted flights at 75 and 100 meters above the surface, and typical flight speed was 8 m/s, though 5 and 7 m/s speeds
were also used in some cases. The longest flight we did was 2 hours 45 minutes. The flight plan was designed using
QGroundControl on the remote controller and transferred to the UAV to perform automated flight. Take-offs and landings
were operated manually.

### 2.2 Laser scanner

The scientific payload (Fig. 3b) for the UAV laser scanning was Riegl VUX-120. The specific feature of the scanner is to
provide three consequential cross-track scan lines across the area of interest at ten-degree intervals. This feature is
implemented using a rotational mirror with tree facets and permits to capture even steep surface shapes in a single flight
pass. The optical configuration gives 100° maximum cross-track field of view, and an effective pulse rate of 1500 kHz. The
laser operates at 1550 nm wavelength and produces 10 mm ranging accuracy and 5 mm precision on targets. Flight
parameters were adjusted to produce 3 cm data spacing for 100 m, and 2 cm for 75 m flight altitudes.

Positioning for the scanner was based on NovAtel CPT7 GNSS-INS with a dual-antenna array by two Harxon HX600A
antennas to provide global reference and accurate attitude information to georeference the laser data (Fig. 4). Positioning
data was recorded at 5 Hz for the GNSS and 400 Hz for the INS data and complemented with simultaneous 5Hz base station
DGNSS data with Trimble R10 receiver.



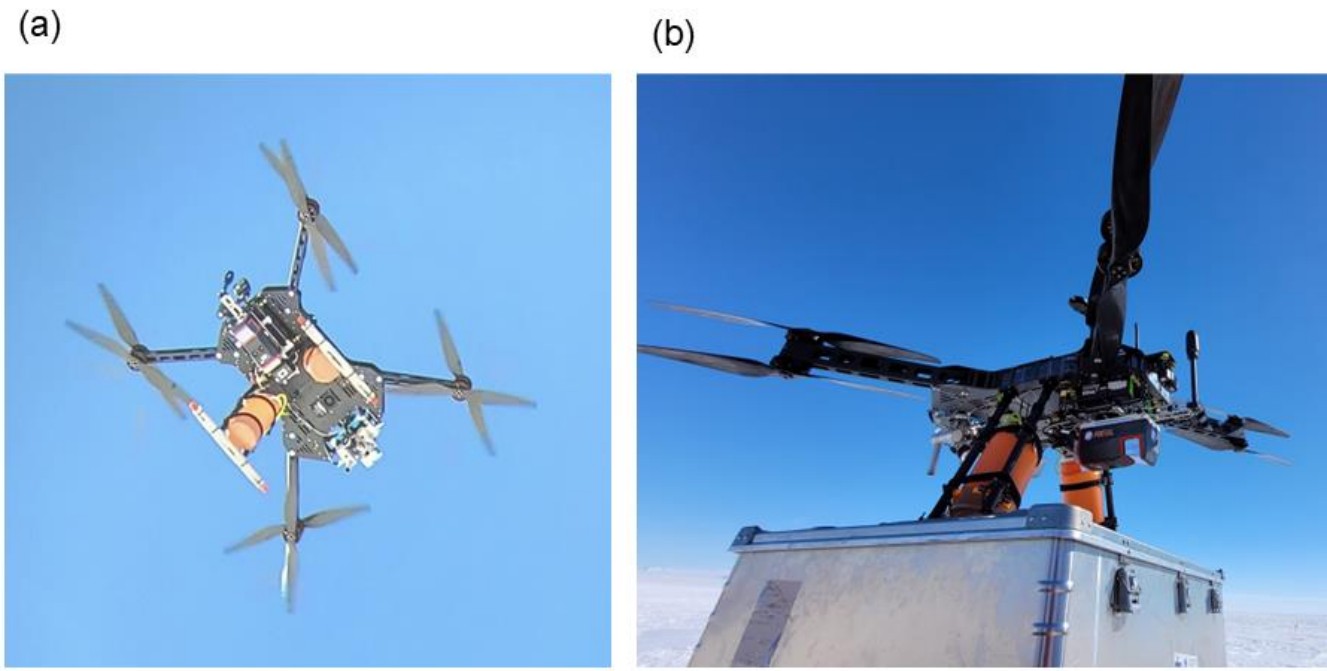

**Figure 3: a) Avartek Boxer and b) Riegl VUX-120 laser scanner mounted on the front of Avartek Boxer drone, GNSS-IMU unit is located right behind the scanner, antenna is seen at the front of the scanner.**


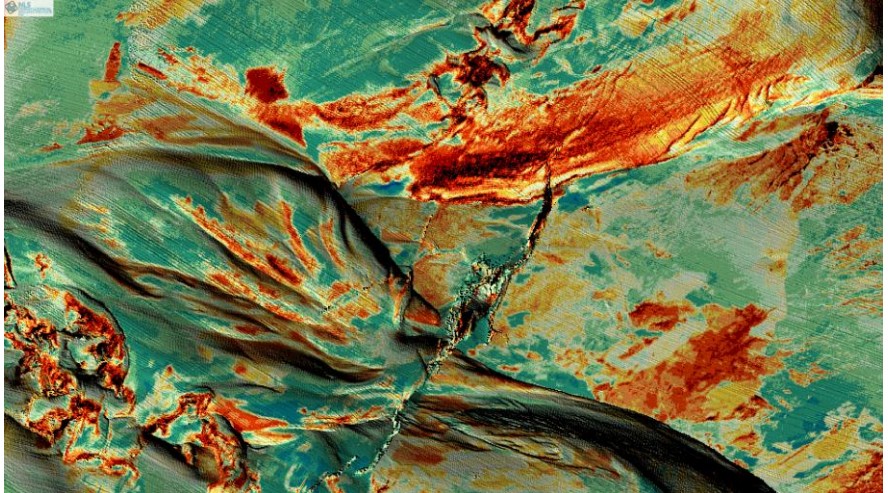

**Figure 4: An example of the laser scanning data at the Riiser-Larsen shelf-ice shows the rugged transition zone with sea ice and wind erosion patterns. Colouring indicates the reflectivity of the surface at 1550 nm.**



## 2.3 Matrice 600 Pro drone

Matrice 600 Pro (DJI) drone was used for radiation measurements (Fig. 5a). The same drone has been used for field measurement in Sodankylä, Finland and therefore, it was proven to be able to operate in winter conditions. The drone uses lithium batteries, and its maximum take-off weight is 15.5 kg. The typical flight speed was 3m/s and flight time was around 15-20 min with one set of batteries, but this varied depending on wind conditions. A typical 84-point flight plan required at least two sets of batteries and took approximately 45 min. Flight altitude was typically 100 meters, but occasionally vertical

radiation profile measurements were made with 100, 85, 65, 50, 40, 30, 20 and 10 m altitudes. Flight plan was similar for each measurement occasion, and the measurement procedure was automated. Gimbal H16 (Gremsy) was mounted on the drone where radiation measurements were attached to be able to level them correctly.

**Figure 5: a) Hyperspectral camera and CM11 in a drone, b) Eppley radiation station, c) removable CM14 station,**

**and d) ASD spectrometer.**



The drone included three GPS sensors. Ground control points were marked with reference flags to be able to georeference the images more accurately. Moreover, accurate GPS location of the reference flags were measured with a precision GPS device (parts from ArduSimple). However, connection problems reduced the accuracy in November and December to approximately 30 cm. In January, problems were fixed, and accuracy was approximately 2 cm. However, a more accurate GPS sensor was located at the laser scanner drone, which data can be combined also with the radiation measurements.

## 2.4 Radiation measurements

Eppley radiation station (Fig. 5b) was installed at the AWS5 site on 19 November 2022 and disassembled on 19 January 2023. The radiation station has two broadband upwelling and downwelling radiative flux sensors observing shortwave and longwave radiation wavelengths. Incoming and outgoing irradiances and albedo are presented for the measurement period in Fig. 6. The instrument has two support legs and the sensors are installed to a bar between them. Installation was made so that shades were minimized on the instrument field-of-view area at noon. The height of the sensors was approximately 120-140 cm above the snow surface depending on whether there was new or drifted snow below the instrument.

The lightweight Kipp & Zonen CM14 radiation station (Fig. 5c) measures both incoming and reflected solar radiation at 340-2200 nm wavelengths. The instrument was mounted on a tripod and azimuthally pointed towards the Sun when measuring at satellite overpass locations. In addition, comparative measurements were made at the AWS5 site with the Eppley radiation station. The height of the sensor was approximately 120-140 cm above the snow surface. The instrument was manually levelled during each setup with a bubble level.

Hyperspectral camera (Rikola Ltd) with 30 channels between 500-900 nm wavelengths was installed on the Matrice 600 Pro drone (Fig. 5a). The channels were chosen to cover the whole wavelength range (Table 3) and to match with optical satellites like Sentinel 2 and Landsat. The hyperspectral camera also included an irradiance sensor and GPS. At AWS5, the covered area was approximately 80 m x 80 m. At the satellite overpass locations, the covered area was 221 m x 85 m which was expanded to 221 m x 119 m when batteries allowed. Measurements were made typically at 36 locations but occasionally also 72 locations in the measurement area depending on the covered area (see Table 2). Drone stopped for each measurement location to allow stable conditions for imaging. Images had approximately 80 % front and side overlap to cover the area so that it is possible to combine a mosaic image for each used wavelength (Fig. 7a). Due to limited possibility to adjust settings for each channel, smaller wavelengths of some images have overexposure problems.





**Figure 6: Incoming (top panel) and outgoing (middle panel) irradiance measured by the Eppley radiation station. Albedo (bottom panel), calculated as the ratio of incoming to outgoing irradiance. Data corresponding to solar zenith angles greater than 80° have been excluded to minimize potential biases due to low sun angles.**




**Table 3: Hyperspectral camera channels (nm).**

| 500-599 nm | 600-699 nm | 700-799 nm | 800-899 nm |
|---|---|---|---|
| 501.001 | 605.523 | 725.086 | 802.533 |
| 559.077 | 618.58 | 738.497 | 815.75 |
| 579.612 | 620.326 | 751.467 | 829.017 |
| 592.534 | 632.582 | 764.489 | 831.676 |
| | 645.806 | 790.276 | 841.0 |
| | 657.933 | | 842.334 |
| | 659.196 | | 854.362 |
| | 663.73 | | 866.431 |
| | 670.588 | | 867.775 |
| | 671.856 | | 879.889 |
| | | | 881.237 |


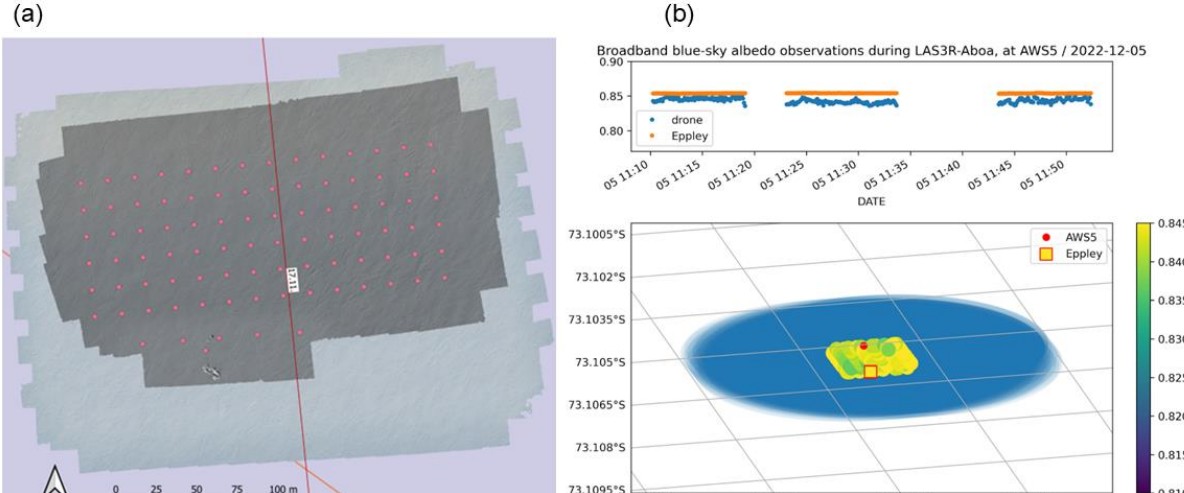

**Figure 7: a) Example of hyperspectral camera data in channel 501 nm on 23 Nov 2022. b) Example of CM11 data on 5 Dec 2022. Top panel presents broadband blue-sky albedo from the drone and Eppley. In the lower panel, spatial**





**variability of CM11 albedo is shown and the blue area presents the field of view of the CM11 instrument at the flight**
**altitude.**

A downward-pointing and gimbal-stabilized broadband Kipp & Zonen CM11 pyranometer (Fig. 5d) with 340-2200 nm wavelengths sensor and 180-degree field of view was also installed to the radiation observation drone. The logger also included a GPS sensor. Radiation was measured continuously when the sensor was powered. Typically, the hyperspectral
camera and pyranometer observations were made simultaneously. However, occasionally the instruments were not successfully started to store data properly and measurements needed to be repeated with an additional flight. Measurement with CM11 on a drone includes more than just snow, such as sky, snowmobiles, and other instruments, due to wide viewing angle of 180 degrees (Fig. 7b).

In addition, a portable ASD Field Spec Pro Jr. spectroradiometer (Fig. 5c) measuring spectral reflectance with 350-2500 nm wavelengths and ~10 nm spectral resolution was used at several locations at the drone measurement area. Locations were chosen visually representing different surface roughness close to measurement of physical surface snow properties. A pistol grip with 8-degree fore optics was mounted on a tripod to stabilize the measurement. In addition, up-wards and down-wards remote cosine receptor (RCR) measurements were made at the snow pit location.

Moreover, traditional RPG photos were also taken from the drone measurement area with the original camera of Mavic 2 Pro (DJI) drone.

## 2.5 Snow pit measurements

A total of 25 snow pits were made at AWS5 and satellite overpass locations during the campaign, excepting the first day on site when the Eppley radiation station was installed instead of the snow pit measurements. Snow pit measurements were made in the vicinity of Eppley radiation station or the removable CM14 radiation station, so that the sampled location was different at every measurement occasion. All snow pits were made towards the direction of the Sun, so that the snow pit wall was shaded (Fig. 8a). The main interest was on snow surface properties and therefore the typical depth of the pit was only
approximately 25-30 cm. Measurement methods were similar as described in Leppänen et al. (2016).

Snow stratigraphy with hardness, grain size, grain type and wetness were analysed according to Fierz et al. (2009). The snow pit area was photographed before digging the pit and photos of the profile including layer interfaces marked with toothpicks was made (Fig. 8b). Grains from every layer were photographed against a SEAR crystal card 1-mm reference grid to enable
analysis of post-processed grain size and grain type (Fig. 8c). Temperature was measured from air, surface, and 1, 2, 4, 6, 10, 15 and 20 cm below surface with HI98501 digital thermometer (Hanna Instruments Ltd) (Fig 8d).



Specific surface area (SSA) was measured with IceCube instrument (A2 Photonic Sensors) which measures hemispherical near-infrared reflectance from snow sample surface using 1310 nm wavelength laser and integrating sphere (Figs. 8e-f), to

be converted to SSA with a dedicated software (Gallet et al., 2009; Zuanon, 2013). This was the first time when the IceCube instrument was used for SSA observations in the vicinity of Aboa station. The measurement was made from the surface and every layer thicker than 2 cm, and for the thick layers, the measurement was repeated approximately every 5 cm.

Density was measured every 3 cm, with a 3 cm high rectangular sampler with a volume of 100 cm$^3$ (Fig. 8g). These snow

samples were weighed with a digital scale (KERN EMB 500-1, KERN & SOHN Gmbh), and the weight of the empty sampler was also recorded. Moreover, liquid water content and density were measured with the Snow Fork instrument (Insinööritoimisto Toikka Oy) from the surface and every 10 cm (Sihvola and Tiuri, 1986). The surface measurements were made holding the instrument vertically so that the measurement covers the topmost 7 cm (Fig. 8h). In the other measurements, the instrument was positioned horizontally to the snowpack.


In addition to the snow pit, spatial variability of surface snow properties was measured at the drone measurement area by repeating the topmost measurements at five locations close to reference flags of the Matrice 900 Pro drone. For those locations, air temperature, surface temperature, grain size and type, SSA, density, liquid water content and density with Snow Fork was measured. Additionally, photos were taken from all measurement locations.


The snowpack in the region consisted mostly of rounded grains (Fig. 9). However, new precipitation particles were deposited on top after a snowfall on the second measurement day in November. Additionally, some ice or crust layers were occasionally detected. Average SSA was 29.4 m$^2$/kg and SSA values varied between 14-52 m$^2$/kg (Fig. 10). However, SSA values were typically between 20-35 m$^2$/kg. SSA was highest, around 50 m$^2$/kg, for the second measurement day in

November when the observed grain type was precipitation particles. Average density was 410 kg/m$^3$ and density varied between 250-520 kg/m$^3$ (Fig. 11). Values measured from the surface were generally close to the values measured from the profile. However, surface values were more often the smallest or largest values from the day. Average liquid water content was 1.8 % and values varied between 0.8-3.8 % but most typical values lie between 1-2 % (Fig. 12). Average surface temperature was -4.6 ℃ and it varied from -8.1 ℃ to +0.2 ℃. Highest surface temperature was not indicating the highest

liquid water content.

SSA as a function of density is shown in Fig. 13. The scatterplot shows that SSA and density are not tightly correlated in the surface snow samples, indicating that surface snow SSA is influenced by grain size, grain type, and liquid water content, in addition to density.




**Figure 8: a) Overview of snow pit work, b) example of profile photo, c) example of grain macrophoto, d) temperature measurement, e) snow sampling for IceCube SSA measurement, f) IceCube instrument, g) density measurement, h) SnowFork measurement from the surface.**



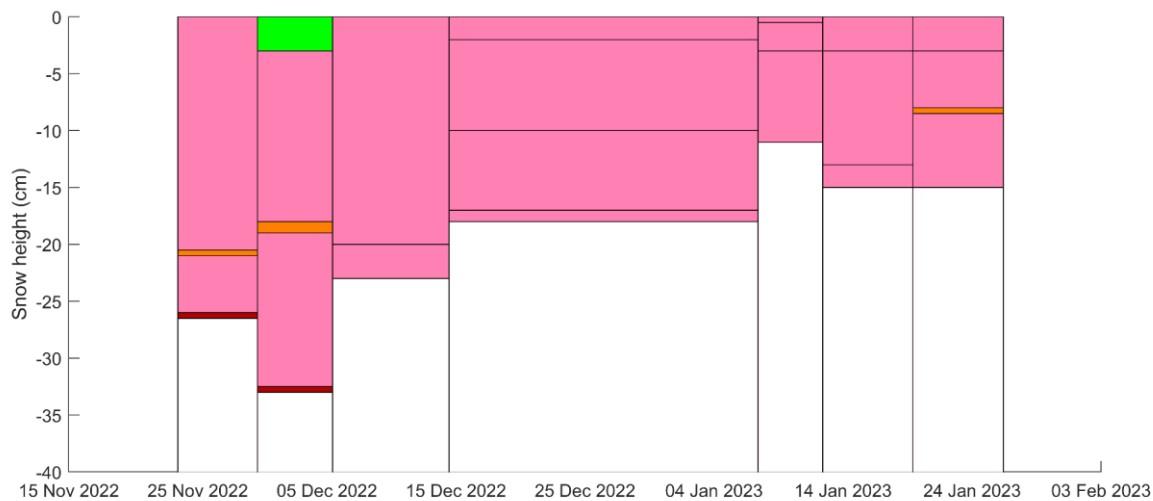

**Figure 9: Stratigraphy for the topmost snowpack. Following grain types were observed: rounded grains (pink), crust or ice layer (red), crust and rounded grains (orange) and precipitation particles (green). Snow depth 0 cm is the surface. Time period for each layer structure is the time between two consecutive snow pits.**

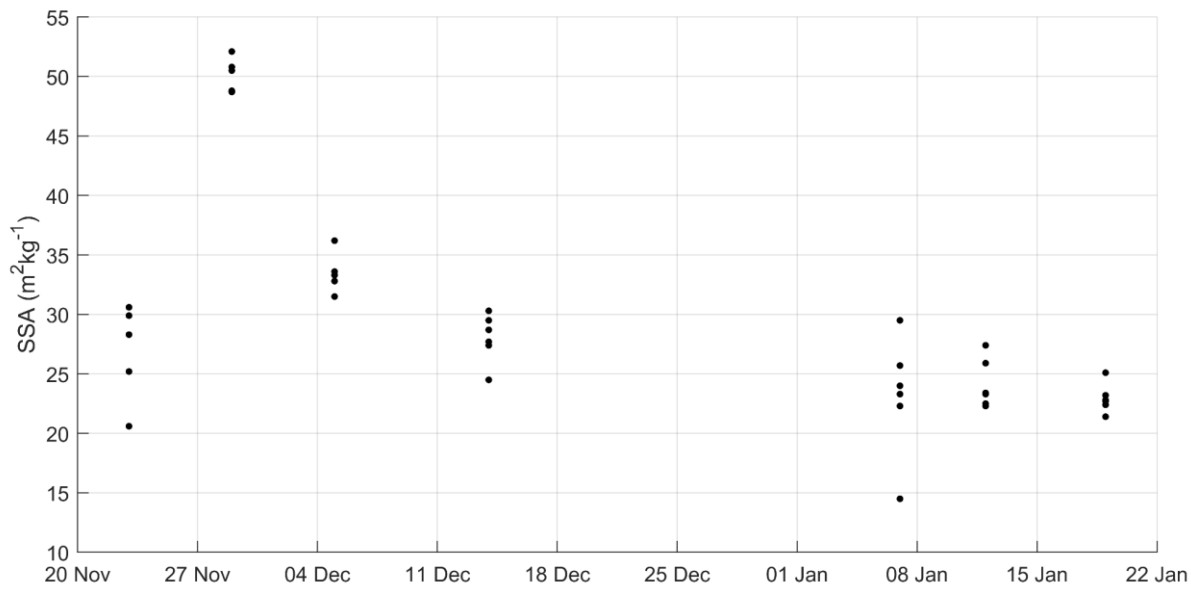

**Figure 10: SSA from the snow surface at the AWS5 site.**



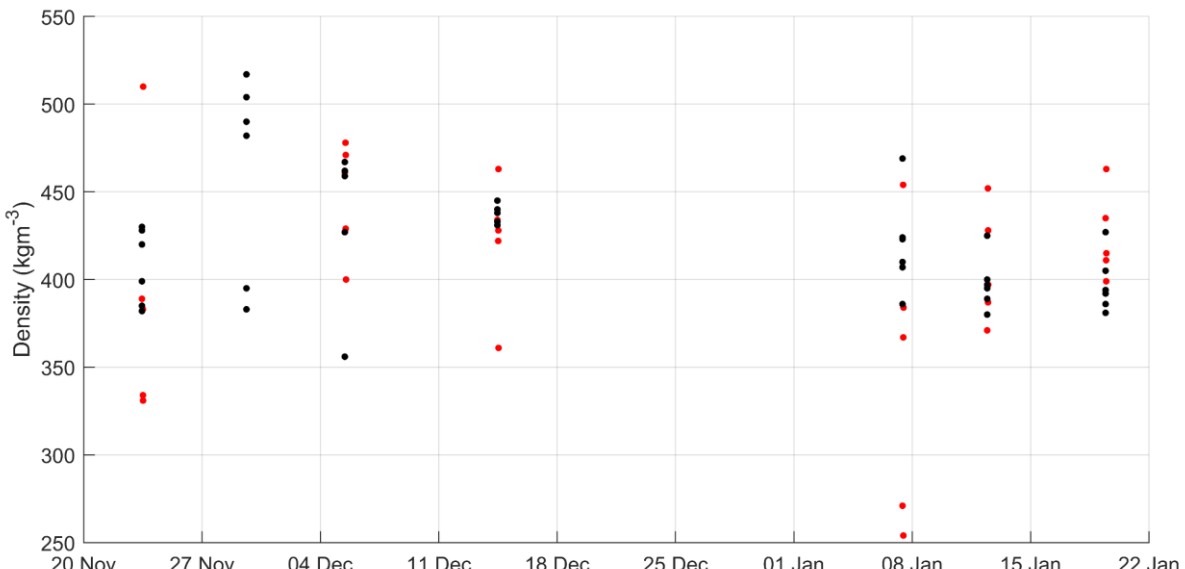

**Figure 11: Snow density profile (black) and surface measurements (red) at the AWS5 site.**


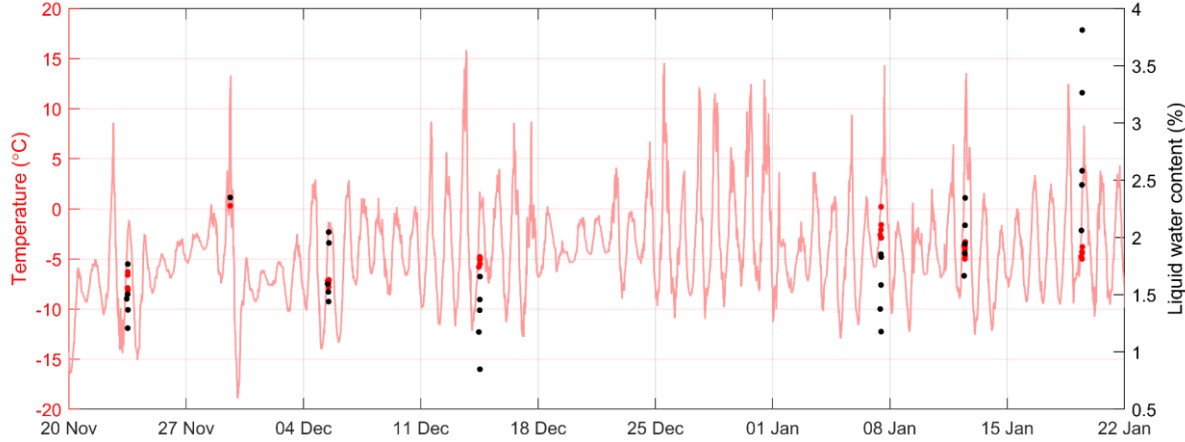

**Figure 12: Liquid water content from snow surface (black), snow surface temperature (red) and air temperature (light red) from an automatic weather station at the AWS5 site.**



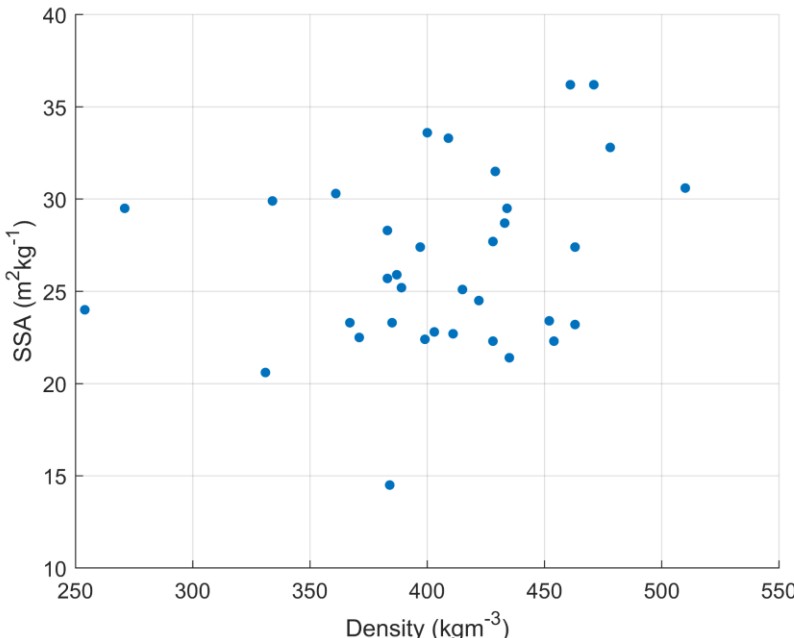


**Figure 13: Scatterplot between SSA and density of surface measurements.**

### 2.5 AWS5 meteorological data

The automatic weather station equipped to continuously monitor key meteorological variables is situated at the AWS5 field site and maintained by FINNARP. Sensor distance to the snow surface is measured with the SR50A sensor (Campbell

Scientific). Air temperature and relative humidity are measured using the HMP110 sensor (Vaisala). Air pressure is measured with the PTB100 sensor (Vaisala) and wind speed is measured using the Heavy Duty Wind Monitor HD-Alpine Model 05108-45 sensor (R&M Young). Surface albedo is determined via upward and downward facing CNR4 pyranometers (Kipp & Zonen). All sensors are mounted on a mast structure. The exact sensor height varies annually due to changes in surface elevation resulting from snow accumulation and ablation. Due to location in the ablation zone, the weather station is

lifted regularly keeping the sensors above the snow surface. Air temperature and sensor distance to the snow surface during the field campaign are presented in Fig. 14. Jumps in the sensor distance from the snow surface potentially originate from ice or snow accumulation to the sensor or wind drift to the measured area.



**Figure 14: a) Air temperature and b) sensor distance from snow surface during the season 2022-23. Measurement**
**days are marked with vertical lines.**

## 3 Data availability

Snow pit data, AWS5 meteorological data, Eppley and CM14 albedo data, and RGB drone mosaics will be available with
DOI 10.57707/fmi-b2share.900cf0c74c1c45efb396b910a841bb3c.

## 4 Conclusion

Surface roughness is one of the most important parameters affecting retrieval of satellite data from Antarctica but it is still
studied little and typically only over small areas. This paper presents a description of a data set collected during Antarctic
summer 2022-23 from the AWS5 station site and IceSat-2 and CryoSat-2 overpass locations in the vicinity of Finnish
Antarctic station Aboa in Dronning Maud Land. In addition to the long-term automatic weather observations at AWS5 site,
the measurement campaign included detailed measurements of snow surface roughness, reflectance, and physical snow

properties. Drone-based observations enabled covering extent area with laser scanner, hyperspectral camera and pyranometer. Spatial variability of snow surface properties was measured from five locations in addition to the snow pit. Total eight sets of data were collected from the AWS5 site and 16 data sets in the vicinity of Aboa station in satellite overpass locations. The presented results in this manuscript are from the AWS5 site, where seasonal snow evolution was monitored. The collected data can be used for improving satellite observations as well as for understanding better connection between albedo, surface roughness, and physical snow properties.

## Acknowledgements

The work has been supported by Finnish Antarctic Research Programme (FINNARP), Research Council of Finland LAS3R project (335986 and 335987), and Next Generation EU Hydro-RI-Platform (Next Generation EU, 246162), Research Council of Finland Digital Waters Flagship (359249), Research Council of Finland Scan4est (346382), the CHARTER project funded under the EU Horizon 2020 Research and Innovation Program (869471). AI was used for language check and creating some of the plotting codes.

## Author contribution

LL, AK and AR participated in the field measurements. LL prepared the manuscript with contributions from all co-authors.

## Competing interests

The authors declare that they have no conflict of interest.

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
