# Peer review of "Measurements on physical snow properties in Dronning Maud Land, Antarctica"

_EGUsphere, 2025_

## Author Comment (AC1)

**General comments:**

This manuscript reports on the field activities of the Finnish Antarctic Research Programme (FINNARP) 2022 expedition, conducted during the 2022–2023 summer season at the Finnish Aboa station in Dronning Maud Land, Western Antarctica. The measurements included continuous meteorological observations at the automatic weather station (AWS), snow pit studies, ground-based and drone-based radiation measurements, and snow surface roughness observations using a laser scanner both from a drone and on the ground. Drone-based measurements using a laser scanner, a hyperspectral camera, and a pyranometer were performed in synchronization with overpasses of the IceSat-2 and CryoSat-2 satellites. The manuscript clearly describes the activities carried out.

I understand that this type of field report by Antarctic research programs is often published as a technical report by the respective programs or institutions. In order for this manuscript to be published in GI, it would be better to highlight new insights related to novel instrumentation, improved observation methods, or innovative combinations of existing techniques. I feel that the current manuscript lacks this aspect. Snow and meteorological observations have been conducted around this station in the past. Is drone-based observation a new feature of this expedition?

Additionally, the authors should clearly state the scientific purpose of this observation program and indicate whether sufficient data were obtained to achieve that purpose. These points are not adequately addressed in the current Introduction and Conclusion.

I suggest that the editor request a major revision of the manuscript from the authors.

*Thank you for the detailed comments which helped us to improve the paper.*

*Drone based observations are the new feature in the data set and it is now emphasized better in the text. In addition, innovative combination of existing techniques is highlighted better. Scientific purpose is also added to the introduction and text about successful data collection is added to the conclusions.*

**Specific comments:**

L29-37: "Aboa is located … (to the end of this paragraph)" This part describes about field measurements. So, these sentences should be moved in Chapter 2.

*Text moved there*

L76: "so that" I don't understand the causal relationship before and after this term.

*Changed to "and"*

L106-107: "Some diurnal variations were found with the lowest albedos being in the mornings and in the evenings" This is opposite property to the general solar zenith angle dependence of albedo. Please explain the possible cause.

*That sentence is mentioned in Rasmus, K.: Field measurements of the total and spectral albedo of snow and ice in Dronning Maud Land, Antarctica, Geophysica, 42(1-2), 17-34, 2006. It is explained by Rasmus as "When surface roughness, such as sastrugi, is present, the diurnal variation of the albedo becomes asymmetrical with regard to the highest solar elevation. At low solar elevation angles when the sun shines from behind the sastrugi, the surface can appear relatively dark producing a low albedo.". Related text added as "Some diurnal variations were found with the lowest albedos being in*

*the mornings and in the evenings since the sun shines from behind the sastrugi and the surface can appear relatively dark.".*

L118: "Aim of this paper is to describe the data set collected during the FINNARP 2022 expedition…" Please describe the scientific purpose of this expedition. In particular, it would be useful to compare the results with past observations and explain what is new and what should be continued.

*Text added "Aim of the expedition was to characterize and quantify the impact of surface roughness on surface albedo and altimetry-based retrieval of snow elevation over Antarctica. The objective was gathering a unique data set to be analysed for surface roughness scale dependence and directionality."*

*Text added "Main purpose of the field campaign during the FINNARP 2022 expedition was to collect drone-based data on snow surface roughness and albedo. Drone-based observations on surface roughness have not been made before in the area and drone-based albedo observations were also more extensive than earlier ones. Spatial coverage was also wide due to measurements at the satellite overpass locations."*

L121: "2 Field measurements" I suggest adding a table that lists the instruments used for this observation, along with their basic specifications.

*Table added.*

L147: "QGroundControl" It should be added a reference or a description that explains what this product is.

*Better explanation is added as "QGroundControl UVS controller software (https://qgroundcontrol.com/)".*

L191: "… Fig. 6." Please explain why some albedos are beyond 1.0.

*In our analysis, these cases correspond to broken-cloud conditions where concurrent enhancement or inhibition of incoming solar flux by cloud shadowing of the direct radiation path and e.g. cloud side reflections combine with alteration of reflected flux by moving cloud shadows in the pyranometer FOV, creating conditions where the measured apparent albedo of fine-grained dry snow may variably exceed 1.0 or be suppressed to 0.8.*

*The measurement itself is thus intact. But, as this is an effect of the measurement conditions and flux aggregation across the pyranometer FOV's rather than any material change in the snow, we propose to revise the figure 6 and text so that the cases (Dec 18 and Jan 1) are filtered, but their presence and effect is explained in the associated text as above. For the reviewer's interest, we enclose a zoomed-in figure of the case on Dec 18 – the high variability in both fluxes is apparent, and being temporally located around mid-day, the possibility of the effect ensuing from e.g. dome riming may be excluded (also note the absence of same effect on the following day). The highly variable effects of broken clouds on the radiation field at ground level are established in the literature. See e.g. Mol, W. and van Heerwaarden, C.: Mechanisms of surface solar irradiance variability under broken clouds, Atmos. Chem. Phys., 25, 4419–4441, https://doi.org/10.5194/acp-25-4419-2025, 2025.*

[Figure]

L222: "Example of hyperspectral camera data" Please indicate what kind of physical quantities is 'Example'?

*Changed as "Example of hyperspectral camera reflectance data"*

L235: "a portable ASD Field Spec Pro Jr. spectroradiometer" I suggest adding a figure showing examples of spectral albedo if it is important for the purpose of this research program.

*We propose the inclusion of the following example illustration of spectral albedo in range 300-2200 nm, as measured by the ASD using an alternation of RCR foreoptic pointing up- and downward on Jan 5, 2023, a clear-sky day. For assessing the quality of the measurement, we simulated black-sky snow albedo with TARTES (Picard and Libois, 2024) and inputs consistent with that day's observed conditions (SSA=18 m2/kg, density=260kg/m3, SZA of 51 deg., direct illumination, climatological impurities of 1.5 ng/g in the surface layer). A commonly quoted 5% relative measurement uncertainty envelope is estimated for the ASD measurement, consisting of calibration uncertainty, the difficulty of establishing a stable thermal regime for the instrument in the harsh conditions, and from the imperfect cosine response of the RCR foreoptic.*

[Figure]

*The agreement between measured and simulated spectral snow albedo is good in the NIR wavelengths where grain size is most significant and e.g. impurities play a minor role. In the SWIR, it is likely that imperfect leveling of the instrument and/or sloping terrain affect the ASD measurement – slope effects are known to affect the SWIR region over snow much more than VIS or NIR (Picard et al., 2016). The shortest wavelengths also show differences above the uncertainty envelope, potentially being caused by a combination of RCR response imperfections and even operator presence if visible in the upward hemisphere during the near-UV measurement which is sensitive to sky blockage. The revised text shall reflect these considerations.*

*Picard, G., Libois, Q., Arnaud, L., Verin, G., and Dumont, M.: Development and calibration of an automatic spectral albedometer to estimate near-surface snow SSA time series, The Cryosphere, 10, 1297–1316, https://doi.org/10.5194/tc-10-1297-2016, 2016.*

*Picard, G., & Libois, Q. (2024). Simulation of snow albedo and solar irradiance profile with the Two-streAm Radiative TransfEr in Snow (TARTES) v2. 0 model. Geoscientific Model Development, 17(24), 8927-8953.*

L254: "SEAR" I don't know if this is a commonly used abbreviation. It would be helpful to explain it.

*SEAR is manufacturer of the crystal card, changed as "a crystal card (SEAR)"*

L288-289: "indicating that surface snow SSA is influenced by grain size, grain type, and liquid water content, in addition to density." I think that there is not shown sufficient evidence to make such a definitive statement.

*We agree and this part was removed as other reviewer suggested.*

L295: In Figure 9, the observed snow stratigraphy appears to have remained stable during the period indicated by the vertical lines. The results of each measurement should be shown in a bar graph for the days on which the observations were made.

*Figure updated accordingly. The stratigraphy is marked for two days to make the bars a bit wider so that the figure is easier to read.*

L314-318: I suggest adding a table summarizing the AWS sensors and their specifications. It would also be helpful for readers to show the picture of the AWS.

*Sensors are added to new Table 3 about the instrumentation. Photo of the station is added*.

L321-322: "Jumps in the sensor distance from the snow surface potentially originate from ice or snow accumulation to the sensor or wind drift to the measured area." Please briefly describe what quality control (QC) have been (or will be) performed, or cite references if they have already been published. It would be helpful to have a summary explanation of QC for other AWS observation data as well.

*In figure 14 (15), if sensor distance from snow surface values has difference to the average of an hour is more than 15 cm, values are now filtered out. Currently, no other quality control has been made for the data. Related text added*.

L329: "**4 Conclusion**" Please indicate whether sufficient data were obtained to achieve that scientific purpose of this research program.

*Text added "The field campaign provided extensive data set for studying impact of surface roughness derived from the laser scanner data on surface albedo and satellite altimetry -based retrieval of snow elevation."*

L330-331: "Surface roughness is one of the most important parameters affecting retrieval of satellite data from Antarctica but it is still studied little and typically only over small areas." This sentence should be mentioned after explaining the overall summary of the following sentences.

*Changed accordingly*

**Technical corrections:**

L52: Please correct "accumulation is strongest" to "accumulation is most abundant".

*Changed*

L155: "3 cm data spacing" Is it a spatial resolution?

*The text was modified as "Flight parameters were set to produce 3 cm along- and across-track point spacing for 100 m, and 2 cm for 75 m flight altitudes at single flight pass right below the UAV. For the complete point cloud data more dense distribution was achieved due to overlapping of the flight lines.".*

L166: Figure 4: Both a distance scale bar and a color scale bar are needed in the figure.

*Added*

L209: Please correct "smaller" to "shorter".

*Changed*

L254: Please correct "Grains" to "Snow grains".

*Changed*

L255: "...and 1, 2, 4, 6, 10, 15 and 20 cm below surface" Please add "snow at" before the figures of snow depth.

*Added*

L258: "Specific surface area (SSA)" Please add "of snow grains" after "(SSA)".

*Added "of snow"*

L274: Please correct "was" to "were".

*Changed*

L300: Figs. 10, 11, and 13: Uniform units are not used in the figures and text. For example, SSA unit is "$m^2$ $kg^{-1}$" in the figures, but "$m^2/kg$" in the text.

*Text is now similar format as the figures.*

L301: Please correct "SSA from the snow surface" to "SSA measured at the snow surface"

*Changed*

L304: "Figure 11: Snow density profile (black) and surface measurements (red)" This caption is somewhat unclear. Do authors mean the snow density values at the surface (red) and vertical distribution values at subsurface depths (black)?

*Surface measurements (red) mean the distributed measurements from the surface. Changed as "Density measured at the snow pit profile (black) and distributed density measurements from the snow surface (red) at the AWS5 site.".*

L306: "Figure 12: Liquid water content from snow surface (black), snow surface temperature (red) and air temperature (light red)" Please correct "(black)", "(red)" and "(light red)" to "(black dots)", "(red dots)" and "(light red curve)", respectively.

*Changed*

---

## Author Comment (AC2)

**General comments:**

This paper reports in situ measurements conducted during the Finnish Antarctic Research Programme (FINNARP) 2022 expedition at the Finnish Aboa station in the Dronning Maud Land, Western Antarctica. Measured physical properties of the surface atmosphere and near-surface snow are listed together with specifications of the employed instruments. Some example data are also presented. These contents are generally well-documented, although I have identified some issues that should be addressed (see below). Therefore, I have confirmed that this paper can provide valuable insights into major national field campaigns and observational research programs in Antarctica, where performing a field campaign is always challenging. These imply that this paper can fit well with the scope of the journal GI.

However, my overall impression is that this paper is something like a field report, because the data, methods, and results are presented in the same section (Sect. 2), which is not a typical composition of a scientific paper. Since I usually don't follow this journal, I investigated some previous papers published in GI and found that the composition of the authors' previous paper, published in GI (Leppänen et al., 2016, cited in this paper), is almost identical to the present paper. Therefore, I understand that the current composition is permissible for GI.

Based on the above, I suggest that this paper may be considered for potential publication in GI once the authors attend to the following points.

*Thank you for the positive feedback and detailed comments which helped us to improve the manuscript.*

**Specific comments (major)**

L. 26 ~ L. 37: The content of the first paragraph of the introduction section is almost the same as that of the abstract, which gives a redundant impression. I recommend writing more general information related to the argument "Understanding the seasonal evolution of Antarctic snow is essential for interpreting satellite observations and quantifying surface mass balance in Antarctica." in the abstract (L. 12 ~ L. 13) here.

*Sentence moved to the introduction.*

L. 118 ~ 120: Reviews of the previous studies (L. 47 ~ L. 116) are OK. So, why did the authors decide to conduct the FINNARP 2022 expedition? The motivation behind the FINNARP 2022 expedition and the scientific relationship between previous expeditions and FINNARP 2022 are unclear. Please consider describing them.

*Related text added ".Aim of the expedition was to characterize and quantify the impact of surface roughness on surface albedo and altimetry-based retrieval of snow elevation over Antarctica. The objective was gathering a unique data set to be analysed for surface roughness scale dependence and directionality." and "Main purpose of the field campaign during the FINNARP 2022 expedition was to collect innovative combination of drone-based data on snow surface roughness and albedo together with in-situ observations of physical snow properties. Drone-based observations on surface roughness have not been made before in the area and drone-based albedo observations were also more extent than existing ones. Spatial coverage was also wide due to measurements at the satellite overpass locations.".*

L. 282 ~ 285 and Fig. 12: In some liquid water content measurements for the surface snow, the measured values are positive although the simultaneously measured surface temperature is negative. How can we believe the measured liquid water content is accurate? Please explain.

*Temperature was measured typically from the shadow and liquid water content measurement was exposed to solar radiation, which might cause positive liquid water content values. In addition, snow surface temperature was measured on top of the snow and vertical measurement of liquid water content contains snow from topmost ~7 cm, where snow can be more wet due to earlier air temperature fluctuations. However, accuracy of the SnowFork instrument needs consideration, which is noticed earlier by comparing SnowFork density measurements with density cutter measurements in unpublished study.*

L 287 ~ 289 and Fig. 13: The purpose of this part is unclear. If the authors want to retain the content, the discussion should be much more in-depth.

*We agree that the discussion was not deep enough, and we decided to remove this part.*

L. 329 ~ 340: Although snow surface roughness is highlighted in the conclusion section, I could not find a figure showing measured snow surface roughness quantitatively. Please reformulate the conclusion section.

*Conclusion is modified, information on surface roughness derivation from laser scanner data is added.*

**Specific comments (minor)**

L. 51 ~ L. 54: The quantitative information here can be more informative if the authors indicate the study period of Reijmer and Broeke (2003).

*Time period of used AWS5 observations is added, 1998-2001.*

L. 71: What do the authors mean by "fine layer structure"? A complementary explanation is needed.

*Changed to "detailed layer structure"*

L. 76: For "grain size", it is necessary to specify whether it is geometric grain size or optically equivalent (optical) grain size.

*Added "traditional grain size" which means largest extent of an average grain*

L. 143, "12 m/s wind speed": What is the measurement level for this wind speed value? Please indicate here.

*Wind speed is measured on ground level, related text added.*

L. 187 ~ L. 194: Please indicate whether the radiation sensors are ventilated or not.

*The radiation sensors were not ventilated due to the power demand being too large for the battery-operated station. The field crew revisited the AWS5 site typically once a week, with a longer break around Christmas and New Year, and ensured that the domes were rime-free. Beyond that, however, it is possible that intermittent riming particularly at dawn/dusk conditions and nighttime is possible. Limiting SZA in analyzable data (e.g. Figure 6) effectively removes most potentially contaminated observations; the few remaining cases are now analyzed in detail (see below). Text is revised to note the lack of ventilation and the regularity of maintenance visits.*

L. 189 ~ L. 190, "shortwave and longwave radiation wavelengths": Please indicate the wavelength range quantitatively.

*Revised to "shortwave (280-2800 nm) and longwave (4-50 μm)".*

Figure 6: Albedo data contains some obvious outliers (albedo > 1.0). These outliers and simultaneous downward and upward shortwave radiations should be masked. Another option is to lower the threshold value for the solar zenith angle (currently set at 80°).

*In our analysis, these cases correspond to broken-cloud conditions where concurrent enhancement or inhibition of incoming solar flux by cloud shadowing of the direct radiation path and e.g. cloud side reflections combine with alteration of reflected flux by moving cloud shadows in the pyranometer FOV, creating conditions where the measured apparent albedo of fine-grained dry snow may variably exceed 1.0 or be suppressed to 0.8.*

*The measurement itself is thus intact. But, as this is an effect of the measurement conditions and flux aggregation across the pyranometer FOV's rather than any material change in the snow, we propose to revise the figure 6 and text so that the cases (Dec 18 and Jan 1) are filtered, but their presence and effect is explained in the associated text as above. For the reviewer's interest, we enclose a zoomed-in figure of the case on Dec 18 – the high variability in both fluxes is apparent, and being temporally located around mid-day, the possibility of the effect ensuing from e.g. dome riming may be excluded (also note the absence of same effect on the following day). The highly variable effects of broken clouds on the radiation field at ground level are established in the literature. See e.g. Mol, W. and van Heerwaarden, C.: Mechanisms of surface solar irradiance variability under broken clouds, Atmos. Chem. Phys., 25, 4419–4441, https://doi.org/10.5194/acp-25-4419-2025, 2025.*

[Figure]

Figure 11: Levels (depths) of the snow density profile measurements should be indicated in the caption.

*Interval of the density measurements is added to the caption.*

Figure 14: Like the case for albedo (Fig. 6), the presented surface meteorological data contains some obvious errors (spikes), which should be masked in my humble opinion.

*If sensor distance from snow surface values has difference to the average of an hour is more than 15 cm, values are now filtered out. Related text added.*

**Technical corrections**

L. 65: It seems to me that Table 1 is not mentioned in the running text.

*Correct, added*

L. 69 ~ L. 70: Is the description "from the topmost 50 m of the snowpack" correct?

*Yes, radar penetration depth was 50m according to the authors.*

L. 89: "found" is better than "resulted"?

*Changed*

Figure 1: An explanation of the red square in the map for the entire Antarctic ice sheet is needed. Also, please describe the satellite data used to show background topography in this figure.

*Explanation about the red square is added. Background is combined from optical satellite photos provided by USGS, which information is now better visible in the figure bottom left corner.*

Figure 2: Please consider indicating the date when this picture was taken.

*Picture was taken 23.11.2022, which is now added to the caption.*

Table 2: When Table 2 is first referred to in the running text (L. 127), CM11 has not yet been introduced. Therefore, in the caption, a detailed description of CM11 should be provided.

*"CM11 pyranometer" added to the table*

L. 150, "(Fig. 3b)": It is better to indicate it at the end of this sentence, in my humble opinion.

*Changed*

Figure 5a: Same as the comment on Table 2. Figure 5 is referred to in the running text before the explanation of CM11 is provided.

*"CM11 pyranometer" added*

L. 227: "Fig. 5d" -> "Fig. 5a"?

*Changed*

Figure 6: "shortwave" must be indicated in the caption.

*Added*

Figure 7: Time zone should be indicated in the upper panel of Fig. 7b.

*Time zone is UTC time and it is added to the caption.*

L. 235: "(Fig. 5c)" -> "(Fig. 5d)"

*Changed*

Figure 8g: It seems to me that the cross section of the snow sample in the density cutter is not flat (this is an example of measurement failure). I recommend replacing the figure if possible.

*Figure changed to better one*

L. 252, "grain size": Please specify whether it is geometric snow grain size or optical snow grain size.

*Added "traditional grain size"*

L. 254, "SEAR": It seems to me that its definition is missing in the running text.

*SEAR is manufacturer and therefore moved to the parenthesis.*

Figure 12 caption: "snow surface temperature (red) and air temperature (light red)" -> something like "snow surface temperature (red circles) and air temperature (light red solid line)"

*Changed*

---

## Author Response (AR2)

**Response to the Associate editor decision**

New chapter "2.6. Novel methodological aspects" is added to show novelty and innovativeness of the study. To increase the instrumental research component and new innovations, description of processing point clouds from laser scanner data is added to the chapter "2.2 Laser scanner" and description of hyperspectral camera data processing is added to the chapter "2.4 Radiation measurements".